# Challenges with pediatric surgical financing and universal health coverage in Guatemala: A qualitative analysis

**Kelsey R. Landrum** [1]*, **Bria J. Hall** [2], **Emily R. Smith** [1,3], **Walter Flores** [4], **Randall Lou-Meda** [5], **Henry E. Rice** [1,2]

**1** Duke Global Health Institute, Durham, North Carolina, United States of America, **2** Department of Surgery, Duke University Medical Center, Durham, North Carolina, United States of America, **3** Robbins College of Health and Human Sciences, Baylor University, Waco, Texas, United States of America, **4** Centro De Estudios Para La Equidad y Gobernanza En Los Sistema De Salud, Guatemala City, Guatemala, **5** Department of Pediatrics, Roosevelt Hospital, Guatemala City, Guatemala

* klandrum@email.unc.edu

## Abstract

The financing of surgical care for children in low- and middle-income countries (LMICs) remains challenging and may restrict adherence to universal health coverage (UHC) frameworks. Our aims were to describe Guatemala's national pediatric surgical financing structure, to identify financing challenges, and to develop recommendations to improve the financing of surgical care for children. We conducted a qualitative study of the financing of surgical care for children in Guatemala's public health system with key informant interviews (n = 20) with experts in the medical, financial, and political health sectors. We used this data to generate recommendations to improve surgical care financing for children. We identified several systemic challenges to the financing of surgical care for children, including passive purchasing structures, complex political contexts, health system fragmentation, widespread use of informal fees for surgical services, and lack of earmarked funding for surgical care. Patient and provider challenges include lack of provider input in non-personnel funding decisions, and patients functioning as both financing agents and beneficiaries in the same financing stream. Key recommendations include reducing health finance system fragmentation through resource pooling, increasing earmarked funding for surgical care with quantifiable outcome measures, engagement with clinical providers in non-personnel budgetary decision-making, and use of innovative financing instruments such as resource pooling. Surgical financing for children in Guatemala requires substantial remodeling to increase access to timely, affordable, and safe surgical care and improve alignment with Guatemala's UHC scheme.

## Introduction

The provision of surgical care remains challenging around the world, particularly for children in low- and middle-income countries (LMICs) [1]. Approximately 11–15% of children in

---

**Data Availability Statement:** The datasets used and/or analyzed during the current study are available from the corresponding author and IRB

Chair on reasonable request. Please contact klandrum@email.unc.edu and walter.lee@duke.edu for data availability questions.

**Funding:** This work was supported by a student development grant given to the lead author (KRL) by the Duke Global Health Institute at Duke University. The funding source did not have involvement in study design, data collection, data analysis, or manuscript preparation. The authors are responsible for the contents of this research article, which do not represent official views of the Duke Global Health Institute or Duke University. No funders played a role in study design, data collection and analysis, manuscript preparation, or the decision to publish.

**Competing interests:** The authors declare that they have no competing interests.

LMICs requiring surgical care by age 15 [2–9]. Despite many areas of surgical care being cost-effective, surgical care financing remains a barrier to timely, safe, and high quality surgical care in many LMICs [10]. An estimated 33 million people are still at risk for catastrophic health expenditures to obtain surgical care annually [1].

Universal Health Coverage (UHC) frameworks aim to ensure that all population members have access to affordable and timely health care [11,12]. The World Health Organization (WHO) set targets of 80% essential healthcare service coverage and 100% protection from catastrophic and impoverishing expenditures within UHC schemes by 2030 [13,14]. The inclusion of surgical care in UHC systems is critical to reducing global morbidity and mortality due to surgical disease [15–17]. However, surgical care remains poorly integrated within UHC frameworks in many LMICs, with UHC priorities often driven by government spending, political influences, and passive purchasing functions rather than by disease burden, local resources, cost-effectiveness, or availability of developmental assistance [18]. Health finance systems are a critical component of UHC frameworks and accurate data describing how surgery is financed and integrated within UHC schemes remain lacking in many LMICs including Guatemala [19].

This study focuses on pediatric surgical care in the Guatemala, a country with extreme challenges to health care, a large youth population, and high levels of poverty [20–22]. The Guatemalan health system is divided into public and private health sectors. The public sector has three providers: the Ministry of Public Health and Social Assistance (MSPAS), the Guatemalan Institute of Social Security (IGSS), and the *Sanidad Militar* (armed forces healthcare system) and the public health sector insures nearly 75% of Guatemala's population [23]. The MSPAS receives 40% of public health spending, IGSS receives 59%, and the *Sanidad Militar* receives 1% [24].

Guatemala's health spending as a percentage of its Gross Domestic Product (GDP) is 5.8%, which is low compared to rest of Latin America, which has a mean health spending as a percentage of national GDP of 8.59% (2016, excluding high income countries (HICs) [20]. In total, Guatemala's GDP is $78.46 billion USD (2017), and the percent of the country's GDP spent on health is small compared to private health expenditures [20]. Private health expenditures comprised 61.9% of domestic health expenditures, or $149.30 USD per capita in 2016 while public health spending per capita is $93.50 USD [20]. Further, the World Bank (2016) estimated that over half (53.3%) of health expenditures were OOP payments, or $128.70 USD per capita [20]. The OOP spending is often on medications, supplies, and other informal fees to obtain care [23].

The 1996 Peace Accords were critical in building Guatemala's healthcare system, as they led to MSPAS reform. The Guatemalan government started implementing the Integrated Health Care System (SIAS) in 1996 to expand primary healthcare access through private provider contracts [25,26]. The government began decentralizing health care in 1996, leading to a decrease in public health spending and an increase in patient OOP spending for health [23,26]. The 1997 Health Code prioritizes the right to health for all Guatemalan citizens [27].

Our study's objective was to provide an understanding of Guatemala's national surgical financing system for children. We used a qualitative approach, conducting key informant interviews with Guatemalan experts in clinical care, health policy, and health financing and using a grounded, applied thematic analysis approach to analysis. Our specific aims were to describe Guatemala's national pediatric surgical financing structure, to identify financing challenges, and to develop recommendations to improve the financing of surgical care for children.

## Methods

We used a qualitative research methods approach, in which we conducted key informant interviews (KIIs, n = 20) with Guatemalan subject matter experts in surgical care, health policy, and health financing. We used applied thematic analysis and grounded theory approaches to analyze and summarize qualitative data to characterize the financing of pediatric surgical care, to identify challenges to financing of surgical care, and to develop a set of recommendations to improve surgical care for children [28].

### Study setting and subject recruitment

This study was conducted from June-July 2019 in the capital, Guatemala City, where most surgical care is concentrated in two national hospitals (Roosevelt Hospital and San Juan de Dios Hospital) that receive 37% of the public hospital budget [29–32]. Interviews were conducted in key informant workplace settings, such as finance, policymaker, and physician offices and physician break rooms.

We used purposive, snowballing sampling for KIIs, with interview participants approached by introduction from other key informants, telephone, email, or WhatsApp. The first participant was approached via introduction from pre-existing contacts in Guatemala. Inclusion criteria were defined as being an expert in health policy, financing, and/or pediatric surgical care in Guatemala (Table 1). Exclusion criteria were defined as not being a subject matter expert in Guatemalan health policy, financing, and/or pediatric surgical care. The study sample is designed to be a representative sample of policy, finance, and surgical experts who are making decisions about or whom are affected by surgical finance policies and challenges in Guatemala. Patients and families were not interviewed, as this study describes supply-side financing. Therefore, the sample is not designed to be representative of the larger, Guatemalan population accessing surgical care. Sampling was designed to reach thematic saturation, with a minimum sample size of 12 KIIs in alignment with qualitative research standards, and is defined by emergency of no new themes in KIIs [33–35].

### Key informant interview procedures

We developed a semi-structured interview guide (S1 File) through review of national health financing reports, the WHO's health systems building block framework, and the World Bank Group's Universal Health Coverage Study Series (UNICO) survey tool [36–38]. The principal investigator (KL) translated the interview tool from English to Spanish and the research team assessed the guide's appropriateness and scope. The interview guide was adjusted according to emerging themes, allowing for data to be coded thematically and interview questions to be open-ended and theory-driven [39]. Participant recruitment and data collection occurred from June 30-July 31, 2019 in Guatemala City in key informant workplace locations. KL conducted all interviews. All participants answered a series of general financing questions, followed by questions within the participant's area of expertise. Interviews (mean duration: 27 minutes) were conducted in Spanish or English and transcribed and translated by KL and a

**Table 1. In-depth interview participant subject matter expertise area.**

| Role | N (%) |
|---|---|
| Finance Expert | 5 (25.0) |
| Clinician | 13 (65.0) |
| Policy Expert | 2 (10.0) |
| Total | 20 |

second study team member. Data were deidentified using unique identifiers available only to study personnel. Audio recordings were deleted after transcription, translation, and de-identification. There was no compensation for participation.

## Applied thematic analysis

We used the WHO and World Bank Group Monitoring Progress Towards Universal Health Coverage at Country and Global Levels Framework to guide analysis using a health systems approach [13,14]. Qualitative coding used an applied thematic analysis approach and was conducted using NVivo 12 [40,41]. KL created structural codes representing study aims and secondary codes representing emerging themes. The study team adjudicated and finalized the codebook after a second study team member coded 15% of interviews. KL coded all data using the final codebook. Results were reported in frequency of occurrence of each code across interviews. Intercoder agreement was assessed, and results adjudicated to ensure a minimum kappa statistic of at least 0.80 (80% agreement) to assess accuracy of application of the final codebook. All analysis was conducted in line with the Consolidated Criteria for Reporting Qualitative Research (COREQ) (S2 File) [42].

Queries, code reports, matrices, and analytic memos were used to assess key themes for each study goal, co-occurrence of codes, and frequency of key themes. We interpreted results in the context of publicly available financing reports [43]. The research team and content experts reviewed and agreed upon all findings and interpretations. We used all data to develop a set of policy recommendations to improve the financing of surgical care for children.

## Ethical approval

According to the policy activities that constitute research at Duke University and the Roosevelt Hospital, this work was considered exempt from review by the Duke University Medical Center Institutional Review Board (Pro00102137). All methods were carried out in accordance with relevant guidelines and regulations. Verbal, informed consent was obtained from all participants, a process approved during the Duke University Medical Center Institutional Review Board exemption process. All methods were carried out in accordance with relevant guidelines and regulation.

# Results

## Pediatric surgical financing system structure

**Formal and informal financing.** Participants described both formal and informal financing streams for pediatric surgical care (Fig 1). In the formal stream, participants described a passive top-down financing stream for surgical care (Fig 2) as a process in which Congress approves a general budget request made by the Ministry of Health, which provides a budget to national hospitals with financing divided into categories of consumable products, not by medical specialty or disease burden.

> *"Here, they divide it by lines, there is a line for equipment, there is a line for supplies. . .if I need medicines and I do not have money in medicines, but I do have money in equipment, that money cannot be used (Clinician)".*

Participants described complex bottom-up financing streams, including funding requests initially made from a surgeon to the social work department, followed by requests to the

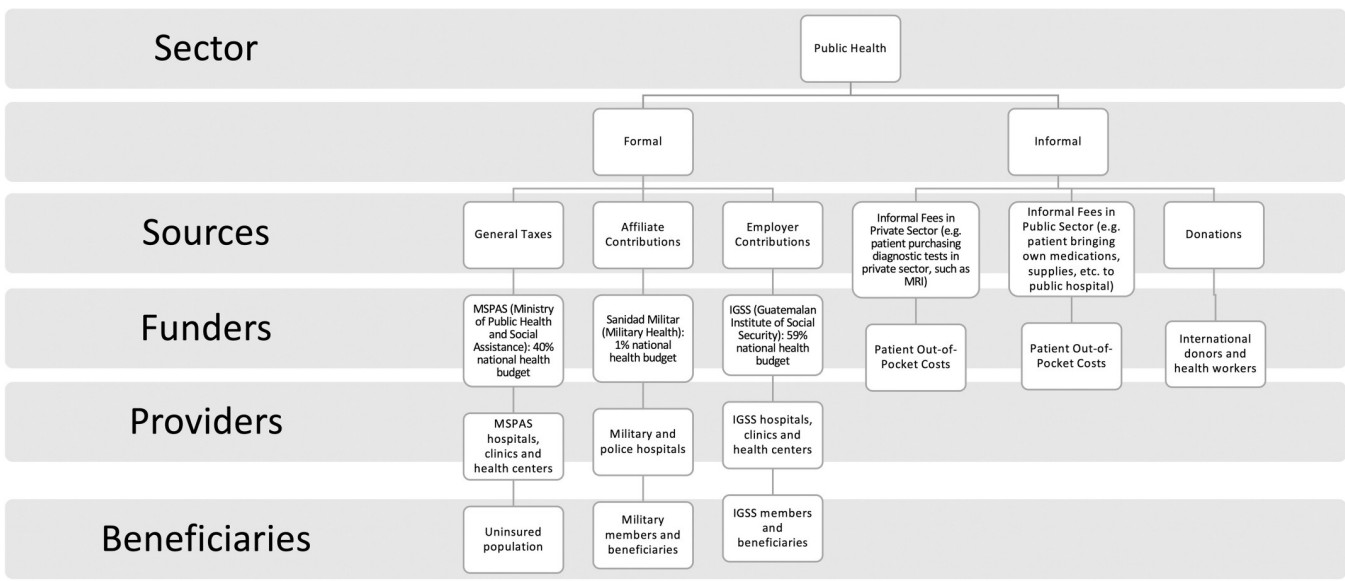

**Fig 1. Financing streams for surgical care in Guatemala.** Source: Adapted with permission from Becerill-Montekio and López-Dávila (2011) to match qualitative findings.

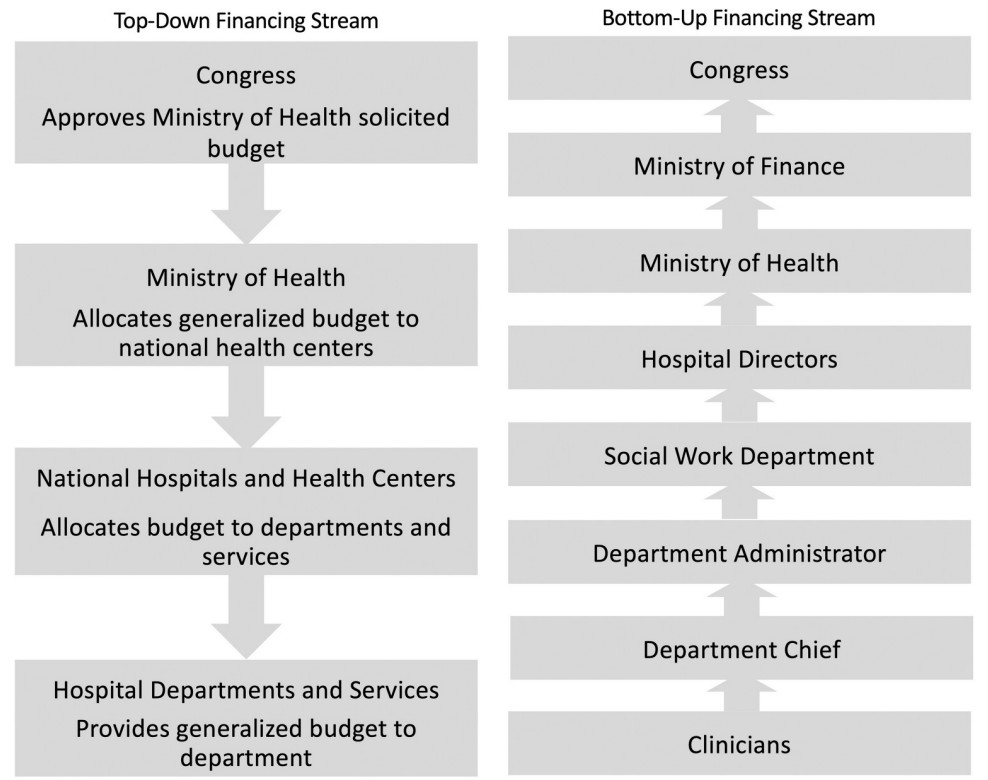

**Fig 2. Top-down and bottom-up pediatric surgical financing streams in Guatemala's public health system.**

department chief and administrator. If the request is large, the department administrator will often present the request to the hospital director. The director submits a budget request to the Ministry of Health, with most appropriations requiring approval by the Ministry of Finance. The surgical department also provides a projection of its annual needs to the hospital administrator, who submits a budget request to the Ministry of Health and Congress.

Participants also identified informal financing for pediatric surgical care. Participants clarified that formal user fees do not occur as they are illegal, but nearly all (n = 16) described informal, OOP payments for pediatric surgical care.

> *"There are no formal fees but there are informal fees quite a lot of them. . .you pay because we really don't have money (Finance Expert)."*

Patients and families incur informal OOP costs through purchasing surgical supplies, such as if the hospital lacks sufficient prosthetics or medications.

> *"If there is something that a patient needs, a medication. . .or prosthesis or whatever that is needed, and the hospital doesn't have it, maybe those cases if the family have money to buy it, they probably would buy it (Clinician)."*

Some respondents felt that many patients cannot afford informal payments and noted a lack of resources in national health facilities as driving informal payment practices. Indirect costs for care, such as transportation, productivity loss, and essential medical materials were described impoverishing and leading to inequitable access to care.

> *"The patients who come spend on their transport, spend on their stay here in the city. . . they stop working. . .it impoverishes them, but the hospital doesn't charge them anything (Clinician)."*

**Financing agents and beneficiaries.**   Primary financing agents include the Ministry of Health, hospital directors, and patients, indicating that patients function as primary financing agents and beneficiaries in the same financing stream. Social work departments were described as intermediary financing agents, often sharing costs of surgical supplies with families.

> *"There is a social service in the hospital that. . .should be able to even go ahead and buy whatever is needed. If that doesn't [work] then the family, if they have money, they will, they will buy some of that (Clinician)."*

A dominant theme was discussion of alternative financing agents, including families, in-kind donations, public-private partnerships, and hospital trusts.

> *"Many times, we [the pediatric surgeons] receive donations, from ourselves. . .to give them, for a study or medication (Clinician)."*

> *"One of the models that is being done, even with pediatric care, is having a foundation that receives some public grants as a contribution, but they do their own fundraising (Finance Expert)."*

Respondents described the *Patronato*, a hospital trust no longer in operation that provided funds directly to families and providers in public hospitals, as an important, unofficial

financing agent for surgical care. One respondent reported that *Patronato* funds supported nearly half of pediatric surgical care at one hospital prior to the government making the trust illegal.

> *"The Patronato will go and buy all, all the things. They will pay for personnel, for nurses. And that's the way it used to work years ago. We even have, the whole pediatric operating rooms, we have six operating rooms. And we were able to, to be, to work with five, five of them when the Patronato was here. . . And when the Patronato left, I mean all that personnel has to go. . . And we had to close those operating rooms. And we went back to three operating rooms (Clinician)."*

Respondents described beneficiaries in the MSPAS system as primarily citizens from lower socioeconomic status and urban populations, or those with *"fewer resources and no private health insurance. It's 80% of the population (Clinician)."*

## Pediatric surgical financing challenges

Dominant challenges (Fig 3) were health system fragmentation and lack of funding. Nearly all participants (n = 19) described a lack of financial resources from the national government, incomplete allocation of solicited budgets, and unorganized hospital spending. Participants

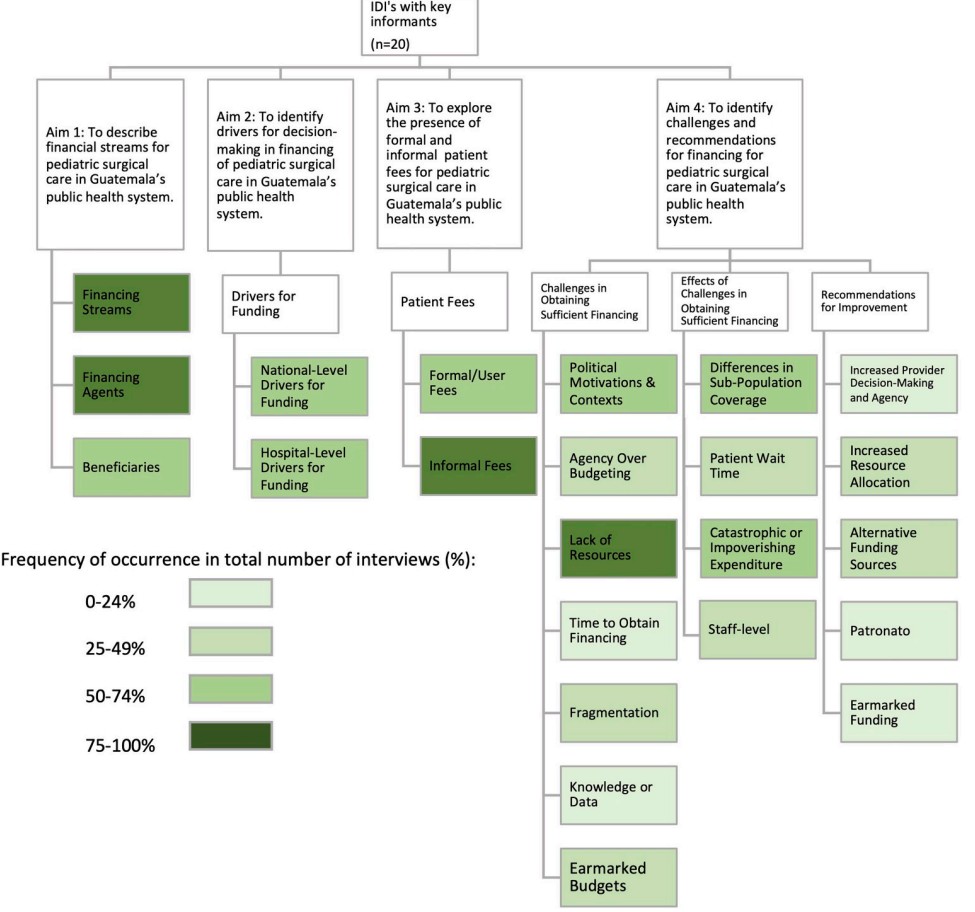

**Fig 3. Dominant themes for pediatric surgical financing in Guatemala.**

described a complex process for providers to request funds for surgical supplies and clinicians reported having to occasionally perform open surgical procedures due to inadequate laparoscopy supplies.

> *"I believe that one of the large problems is a lack of coordination in a lot of the entities that work in favor of the improvement, eh, of health of children (Finance Expert)."*

> *"September, October, the budget runs out and the medicine runs out. And there is no more (Clinician)."*

> *"Second issue. Let's go back to the public health system. In the city, in the city, up to today we have grown enormously in the past ten years and we have only two big university hospitals here." (Policy Expert)*

Lack of earmarked financing for surgical care from the national government was also a dominant theme. Responses (n = 9) indicate that surgical care does not have earmarked financing in the national health system. As such, clinicians assume some of the primary financing responsibilities to obtain supplies.

> *"Department resources come directly from the hospital budget. So, you have to do management, you have to do a lot of complex procedures to get the necessary resources (Clinician)."*

> *"There is no budget allocated for pediatric surgery as such because we belong to the hospital. . . Then it is distributed into everything. . .And the result is that there is no more money left, even for basic things. The antibiotic, amoxicillin, sometimes there is none, and a hospital should not lack of such a simple antibiotic, but there are times when very simple things are missing (Clinician)."*

> *"It is not possible to have only the national programs and the view of the Ministry of Health with this perspective of top-down approach, or only have our vision as the local. . .And this is all the, the topics that we are trying to solve at this point, yeah? Because we have fragmented health systems." (Policy Expert)*

Lack of provider ability to inform non-personnel budget decision making was a dominant theme (n = 6). Clinicians reported not having decision-making authority over how national, hospital, and service-level budgets are apportioned. The lack of provider understanding of how national budgets are operated was consistent with providers' sentiment about not having a voice in budget decision making. Four participants described a lack of knowledge over surgical budgets as a challenge for the health system's surgical financing mechanisms.

> *"We don't, we don't have any, any at all, any decisions in how much the, what budget is going be or how much else we should be getting. I mean, we have no decision on that (Clinician)."*

Half of participants (n = 10) reported inequities in coverage, with children in low socioeconomic status and rural populations having decreased access to care because surgical capacity is concentrated in Guatemala City. Patients with IGSS coverage may purchase additional private insurance to cover the costs of specialized and emergency care.

> *"Multidisciplinary treatment for the pediatric patient outside the city [the capital] is very limited. That's why they have to refer the majority, if not 98%, of neonatal surgical patients, refer*

*them here [to Hospital Roosevelt or San Juan de Dios]. But here it is so full that many of the patients cannot be assisted, and the result is not really the right one." (Clinician)*

*"From the perspective of universal health coverage, we have financial protection gaps and difficulties. . .the people have different difficulties for access or they are considered as very dangerous regions in the municipality so the Ministry of Health don't go inside in these regions because its danger[ous]" (Policy Expert)*

Respondents (n = 7) reported financing challenges at staff and provider levels. One participant described having to choose between patient well-being or penalties for providing care if they do not follow government protocols. Another participant referred to tension among hospital staff due to unorganized budget allocation. Other patient-level financing challenges include patients having delayed release from the hospital. Participants discussed lack of fiscal reform and taxation mechanisms, the need for improved UHC monitoring, and surgery not being seen as an essential part of public health.

*"Because we know that they are not going to comply with their treatments or because they do not have money, then they stay longer than necessary in the hospital. . .it delays their discharge, it limits other patients who need to enter, the rotation of beds is obviously longer. . .We do it for the benefit of one patient and then many times we sacrifice other many patients, which we cannot measure how many we miss (Clinician)."*

**Drivers for pediatric surgical financing.** Nearly all participants (n = 17) described at least one national and/or hospital-level driver of surgical financing. National-level drivers included political contexts, patient volume, projected needs, and historical budgets. Several participants described that financing is influenced by politics and labor unions. Participants described budget making as non-needs based, and over half of participants (n = 13) identified political contexts as barriers to financing.

*"I believe that the budget, eh, has a technical analysis more not an analysis in which they prioritize the needs and therefore the allocation, again, I don't believe it has, in my criteria, a profound level that responds to the specific needs (Finance Expert)."*

Many participants (n = 14) discussed hospital-level drivers for financing. Clinicians have a limited understanding of what motivates their departments' surgical budget and spending, and several discussed lacking input in hospital finance decision-making. Finance experts noted that budget making is based on hospital and political processes and past budgets and lacks needs-based analysis.

*"What is assigned for each place, we do not know (Clinician)."*

*"The truth is it is hard. . .they should do it on the basis of production, but they do it a little on the basis in between production and history, historical series (Clinician)."*

## Key informant recommendations for improving pediatric surgical financing

A dominant recommendation was increasing provider and departmental control over non-personnel budgetary decisions. One clinician described increasing provider independence in

budget allocation while another described how the pediatric surgical department should have its own budget. Participants recommended that drivers for financing should be related to technical analysis and population needs rather than by political motives.

> *"The department would have a certain budget that it could execute directly. . .that we had the ability to run it easily, without all the big process you have to do (Clinician)."*

> *"More than anything, to speed up the purchasing process. To streamline all that bureaucracy and the bunch of laps that you have to do to be able to authorize buying something (Clinician)."*

Several participants (n = 5) suggested alternative financing mechanisms for surgical care, noting that funding comes solely from taxation mechanisms. Participants recommended use of public-private partnerships similar to existing models for specialized pediatric heart and cancer care (*La Unidad Nacional de Oncología Pediátrica* (UNOP) and *La Unidad de Cirugia Cardiovascular de Guatemala* (UNICAR)). Recommendations also included consideration of pooling MSPAS and IGSS resources, and for the government to allow providers to ask patients to pay for a direct portion of health expenses if they are able. Finally, one participant recommended reinstating previously operational hospital trusts, such as the *Patronato*, to support financing of care.

Nearly all participants described lack of financial resources as a barrier to surgical care, and five made recommendations to improve resource allocation. Four participants recommended changing the finance decision making process, with one finance expert explaining how a major health reform is necessary but that political contexts preclude such reform and one participant recommending reducing health system fragmentation. These recommendations suggest mechanisms to better to align budget-making processes with data from needs-assessments.

## Discussion

Our analysis suggests that existing financing streams in Guatemala are complex, inefficient, and pose substantial challenges to the delivery of timely and affordable surgical care for children. There are systemic challenges to the financing of surgical care, including passive purchasing structures, complex political contexts, health system fragmentation, widespread use of informal fees for surgical services, and lack of earmarked funding for surgical care. We also identified patient and provider challenges, including lack of provider input in non-personnel funding decisions, and patients functioning as both financing agents and beneficiaries in the same financing stream.

The use of OOP and informal payments for surgical care remains a major barrier to care [1,13]. Our findings are consistent with literature indicating that health financing in Guatemala relies heavily on OOP spending, with over half of the country's health expenditures in 2018 being OOP payments [20,23,44]. The practice of informal OOP payments remains for pediatric surgical care, as OOP payments can lead to risks of impoverishment among vulnerable populations [23,45–47].

Our review indicated that surgical financing data are limited and difficult to access. The few available reports suggest that national health budgets are based on locations of care and allocated by consumable supplies rather than by medical need or specialty, a finding consistent with key informant description of pediatric surgical funding mechanisms [48,49]. Similarly, data on the extent of informal payments for surgical care are prohibitively challenging to

precisely track, although the World Bank estimated that 38% of Guatemalan citizens were at risk of catastrophic health expenditure for surgical care in 2017 [20].

Our analysis suggests several strategies to improve the financing of surgical care for children, including implementation of evidence-based finance decision making, improved organizational infrastructure, and reduction of informal OOP payments through use of innovative financing mechanisms.

## 1. Evidence-based financing and organizational infrastructure

National and hospital-level financing systems should incorporate objective data to develop surgical financing streams. Current budgetary decisions are developed with limited data on national or local surgical needs, resources, and expenditures, and instead rely on political contexts, historical budgets, and estimated patient volume. This process is consistent with practices across many high-, middle-, and low-income countries [18,50]. We found that neither cost-effectiveness nor local disease burden are major drivers for surgical funding in Guatemala, limiting efficient resource allocation [16,17,51]. Building a culture of evidence-based policy based on local population health needs would markedly advance financing in Guatemala's public health system, including using costs and benefits of surgical interventions through tools such as decision analysis [52–54].

Use of a needs-based financing program requires a functional and bidirectional relationship between health providers and state actors, a need indicated by several participants [55–58]. Increased provider voice in non-personnel budgetary decision making is essential, as the lack of provider input in financing limits front-line knowledge of resource needs for high-quality care. Although provider gatekeeping may lead to inefficiencies and conflicts of interest, the use of clinicians in these processes would provide subject matter expertise to guide financing [59]. Other countries can serve as a model for these practices. For example, Thailand's use of strategic purchasing in its UHC package integrates subject matter expertise with needs-based finance mechanisms. In several LMICs, the integration of a National Surgical, Obstetric and Anesthesia Plan (NSOAP) within National Health Plans (NHPs) has advanced practices to improve surgical financing processes [59–62]. We encourage the use of evidence-based financing systems as well as consideration of a development of a NSOAP in Guatemala to improve financing schemes and alignment with UHC frameworks [1,63].

Developing hospital budgets for surgical care based on local population needs would build efficiency of budgets and decrease uncertainty about financial risks [64]. This top-down approach would improve the efficiency of existing, non-linear finance structures, and address health system fragmentation and inefficient bottom-up financing streams. Evidence-based systems to support budget and pre-payment processes should be integrated with health financing platforms, processes, and feedback mechanisms to track surgical expenditures across health system levels. Allocation of finances based on disease burden requires streamlining and reorganizing health financing infrastructures and payment systems, and requires a linear and easily navigable organizational infrastructure. As a model, Argentina's social security system has a strong financing organization which has increased health system performance and decreased costs [65].

## Informal user fee reduction

If lack of financial protection is analogous to a health system "disease," OOP payments for surgical care are a symptom of a complicated health system condition [66]. Informal payments for surgical care in Guatemala should be replaced by innovative financing mechanisms, such as pooling of alternative, horizontally based resources. Our findings suggest that formal user

fees for surgical care have been replaced by informal user fees [23]. Further complicating our analysis is the lack of quantitative measurement of informal payments. Ironically, this missing data may result in a falsely high, favorable UHC index (measure of OOP expenditures) due to under-estimation of OOP expenditures. However, the use of informal fees to finance pediatric surgical care affects the provision of timely and quality surgical care, further muddling existing unorganized bottom-up supply chains and financing processes.

In some LMIC settings, user fees have been shown to increase facility and provider-level ability to purchase medical supplies and increase health service use [67–69]. However, user fees disproportionately impact lower socioeconomic status populations, contributing to impoverishing and catastrophic health expenditures and undermining UHC philosophies [13,44,70]. Patients paying for care informally have no negotiation power, increasing their financial risk and decreasing their ability to pool risks [45,64,66]. The unregulated presence of informal payments contributes to health inequities, with patients who are able to pay for care having preferential access to care compared to those with fewer assets [44,45]. Informal payments also have adverse national-level economic consequences, as increased OOP expenditures can depress economic development [66]. We advocate for elimination of the practice of OOP spending for surgical care in Guatemala, as the growing body of evidence around the world suggests that user fee reduction can increase health service use, particularly when combined with quality improvement measures [69,71–73].

## Innovative financing mechanisms

Innovative financing mechanisms, such as progressive taxation reform or non-governmental organization (NGO) support, could improve access to pediatric surgical care in Guatemala and compensate for funds lost from removal of informal payments [63]. Taxation opportunities within Guatemala's fiscal space include mobilizing domestic agriculture funds, such as earmarking a proportion of sugar tax revenue towards national health system strengthening [63,74,75]. However, as confirmed by our key informants, tax-based financing reforms would be difficult in Guatemala's current political context [50].

There are several alternative financing resources to offset losses resulting from informal user fee removal. NGOs already support a large portion of health care in Guatemala and may be a valuable source for surgical financing. Approximately 18% of Guatemalan citizens receive health care through NGOs, and public-private partnerships NGO have been successful models for specialized cardiac and cancer care for children in Guatemala [23,76,77]. Pooling donor and government support (a model similar to the Global Fund) may be another mechanism to increase surgical financing [63,78,79]. Such systems can cross-subsidize prepayment for care and spread financial risks across health system components [80–85]. Long-term donor partnerships similar to the Global Vaccine Alliance (GAVI) could create sustainable donor and government support [18,63,86,87]. Finally, pooling of tax-based MSPAS and contributory-based IGSS resources could streamline health system revenues, allowing residual funds to be earmarked for select segments of the health system.

## Study strengths and limitations

Strengths of our study include the PI conducting all interviews to increase internal validity and consistency across interviews. The use of qualitative methods for assessment of complex financing themes which are beyond the scope of solely quantitative or quantitative data. Study limitations include biases related to political contexts possibly having influenced key informant responses. Most participants engaged in discussion of all topics and did not appear limited in their responses. We did not analyze findings by professional role, as overlap in

participant roles and limited representation from policy and finance experts precluded such analysis. Importantly, our analysis was focused on supply-side views of financing challenges, and did not address the important views of families and patients. These analyses are critical to fully contextualize the microeconomic impacts of financing challenges on families and communities. The sample was not representative of the Guatemalan population and may differ based on socioeconomic status. Our interview guides were informed by select themes in prior studies, raising the possibility for selection bias. Finally, triangulation of qualitative and quantitative data was precluded by lack of quantitative data on user fee presence, surgical costs, or surgical budgets in Guatemala. Instead, we chose to analyze our qualitative data within the context of the limited existing quantitative data.

## Conclusions

Pediatric surgical financing in Guatemala faces complex challenges that must be systematically addressed to increase access to timely, affordable, and safe surgical care for children. Without adequate financing, clinicians and administrators are forced to choose between following inefficient financing protocols to provide supplies or using "work-arounds" in existing finance systems. Investment in strong surgical financing is a global public good, and Guatemala should strive to develop a surgical financing system that is in line with UHC frameworks.

Our recommendations for improving surgical financing for children in Guatemala's national public health system are as follows:

1. Finance decision-making at the national and hospital levels should use evidence-based data to align adequate financing streams with local population needs and should prioritize inclusion of pediatric surgical care when defining essential benefits packages.

2. Health financing infrastructures should be strengthened through organization of platforms, processes, pre-payment, and feedback mechanisms at the national and hospital levels.

3. Innovative financing instruments, including pooled donor support, taxation mechanisms, expanded health insurance funding resources, and public-private partnerships, should be used to replace informal user fees.

4. Financing resources should be directed toward horizontal-based health system strengthening to replace OOP expenditures for surgical care.

## Supporting information

**S1 File. Clinician semi-structured in-depth interview guide.**
(PDF)

**S2 File. COREQ (COnsolidated criteria for REporting Qualitative research) Checklist: Challenges with pediatric surgical financing and universal health coverage in Guatemala: A qualitative analysis.** Developed from: Tong A, Sainsbury P, Craig J. Consolidated criteria for reporting qualitative research (COREQ): A 32-item checklist for interviews and focus groups. International Journal for Quality in Health Care. 2007. Volume 19, Number 6: pp. 349–357.
(PDF)

## Acknowledgments

The authors acknowledge la Fundación Desarolla Guatemala (FUNDEGUA), el Centro de Estudios para la Equidad y Gobernanza en los Sistemas de Salud (CEGSS), and la Fundación

para el Niño Enfermo Renal (FUNDANIER) for their academic support and leadership in this endeavor and for their commitment towards improving access to healthcare for children and for Guatemalan citizens. We thank the Duke Global Health Institute for their leadership in improving health care for children worldwide. We also thank Melany Puente and Dr. Osondu Ogbuoji for lending their assistance and academic expertise to this study.

## Author Contributions

**Conceptualization:** Kelsey R. Landrum, Bria J. Hall, Emily R. Smith, Henry E. Rice.

**Data curation:** Kelsey R. Landrum, Walter Flores, Randall Lou-Meda, Henry E. Rice.

**Formal analysis:** Kelsey R. Landrum, Bria J. Hall, Henry E. Rice.

**Funding acquisition:** Kelsey R. Landrum.

**Investigation:** Kelsey R. Landrum, Bria J. Hall, Emily R. Smith, Walter Flores, Randall Lou-Meda, Henry E. Rice.

**Methodology:** Kelsey R. Landrum, Bria J. Hall, Emily R. Smith, Henry E. Rice.

**Project administration:** Kelsey R. Landrum.

**Resources:** Kelsey R. Landrum.

**Software:** Kelsey R. Landrum.

**Supervision:** Emily R. Smith, Henry E. Rice.

**Visualization:** Kelsey R. Landrum.

**Writing – original draft:** Kelsey R. Landrum.

**Writing – review & editing:** Kelsey R. Landrum, Bria J. Hall, Emily R. Smith, Walter Flores, Randall Lou-Meda, Henry E. Rice.

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
