## [Decision Letter · Decision Letter 0]

28 Jan 2022

PGPH-D-21-00647

Challenges with pediatric surgical financing and universal health coverage in Guatemala: A qualitative analysis

Dear Dr. Landrum,

Thank you for submitting your manuscript to PLOS Global Public Health. After careful consideration, we feel that it has merit but does not fully meet PLOS Global Public Health’s publication criteria as it currently stands. Therefore, we invite you to submit a revised version of the manuscript that addresses the points raised during the review process.

We look forward to receiving your revised manuscript.

Kind regards,

Augustine D. Asante

Academic Editor

Journal Requirements:

1. Please provide additional details regarding participant consent. In the ethics statement in the Methods and online submission information, please ensure that you have specified whether vewrbal consent was informed consent.

2. In your Methods section, please provide additional information about the recruitment method for key informants and their demographic details. Please ensure you have provided sufficient details to replicate the analyses such as: 

a) the recruitment/interview date range (month and year), 

b) a description of any inclusion/exclusion criteria that were applied to participant recruitment, 

c) a table of relevant demographic details, 

d) a statement as to whether your sample can be considered representative of a larger population

3. In the online submission form, you indicated that "The datasets used and/or analyzed during the current study are available from the corresponding author on reasonable request.". All PLOS journals now require all data underlying the findings described in their manuscript to be freely available to other researchers, either 1. In a public repository, 2. Within the manuscript itself, or 3. Uploaded as supplementary information.

4. Please change the item type of file "PermissionRequestForm_Figure_1_VBecerril.pdf   " as "Other": 

Additional Editor Comments (if provided):

Additional comments:

1. Rigor - the authors need to demonstrate that rigor was ensured in this study. How was internal validity and reliability ensured and to what extent can the results be generalized? Were there any attempts to triangulate the data?

2. A study of this nature would require substantial insights from policymakers. You interviewed two policy experts but their views are not well reflected in the results. Rather, the results seem to prioritize the views of clinicians - you can't solve a financing issue from a clinical perspective only.

3. The background needs to properly describe the health financing and delivery context in Guatemala - how much does Guatemala spends on health as a proportion of GDP? What proportion of general government expenditure is allocated to health care? What is the health expenditure per capita? How much of total health expenditure is accounted for by out-of-pocket payments? How much do donors put in? How are hospital services financed in Guatemala - is it on a fee-for-service basis? casemix? how is that different from the financing of primary care services? What role does the private sector play in the provision of surgical care in Guatemala? Such details are necessary to put the study in a proper context.

4. Methods - this section needs to be strengthened, especially the recruitment of participants - this is not sufficiently described. The sampling procedure sounds like a snowballing sampling approach, but this was not mentioned. How was the first key informant identified, approached and recruited? Under the 'interview procedures' indicate that the PI conducted all the interviews.

5. You raised the issue of informal payments and the risk of impoverishment - this is important, but I wonder how it can be properly assessed without talking to patients; clinicians views about this may be completely different from what patients experience or think.

6. Reading the paper, I did not get a good sense of how the issue of surgical financing in Guatemala can be addressed in the context of universal health coverage. Earmark funding for surgical care, in my view, is not the way to go; what is important under UHC is for Guatemala to develop a better prepayment system (health insurance) with a comprehensive benefits package that includes key hospital services such as pediatric surgical services. This is what will address OOP health spending especially by the poor and ensure equity. Your discussion can emphasize these issues.

Reviewers' comments:

Reviewer's Responses to Questions

**Comments to the Author**

1. Does this manuscript meet PLOS Global Public Health’s publication criteria? Is the manuscript technically sound, and do the data support the conclusions? The manuscript must describe methodologically and ethically rigorous research with conclusions that are appropriately drawn based on the data presented.

Reviewer #1: Yes

Reviewer #2: Yes

2. Has the statistical analysis been performed appropriately and rigorously?

Reviewer #1: N/A

Reviewer #2: N/A

3. Have the authors made all data underlying the findings in their manuscript fully available (please refer to the Data Availability Statement at the start of the manuscript PDF file)?

Reviewer #1: No

Reviewer #2: Yes

4. Is the manuscript presented in an intelligible fashion and written in standard English?

Reviewer #1: Yes

Reviewer #2: Yes

5. Review Comments to the Author

Reviewer #1: This is a well-written paper on a seemingly important topic. I only have a few issues.

Given time of data collection, I am surprised that KIIs made no reference to Covid 19; neither do the authors e.g. within recommendations. Would some of these issues emerge if there was no Covid?

Abstract - guidelines indicate this should have headings (Introduction, Materials and methods, Results, Discussion, Conclusion)

Sampling was designed to reach data saturation, with a minimum sample size of 12 KIIs. How and why did the researchers get to 20? Did they achieve theme saturation with the 20 KIIs?

If PI is one of the author - line 104 and subsequently, use their actual initials.

Line 88 - give full form of OOP as this is first time mentioned and full form is never given.

Reviewer #2: Review of "Challenges with pediatric surgical financing and universal health coverage in Guatemala: A qualitative analysis"

Comments:

The topic of the paper is well-motivated in the introduction.

I believe that the introduction section is too short, and it does not include several points of the investigation, which would be interesting to read in the first section of the paper. For example, a summary of the context of the research for Guatemala, the methods, and the main findings may be presented in the introduction.

The “Methods” section is too short.

The “Setting” section, or a part of it, may be included in the Introduction to describe the context of the research for Guatemala.

There are some short sections and the material presented in those sections is for some cases belongs to the introduction and for some cases it belongs to a methods section. Please review the organization of the different sections of the paper.

Nevertheless, in the separate sections, I believe that the detail of the method is well-presented. Please put them together into a longer methods section.

I believe that a possible structure of the paper could be: 1) Introduction; 2) Methods; 3) Results; 4) Discussion; 5) Conclusions.

The results, discussion, and conclusion sections are well-presented.

6. PLOS authors have the option to publish the peer review history of their article (what does this mean?). If published, this will include your full peer review and any attached files.

**Do you want your identity to be public for this peer review?** For information about this choice, including consent withdrawal, please see our Privacy Policy.

Reviewer #1: No

Reviewer #2: **Yes: **Szabolcs Blazsek

---

## [Editor Report · Decision Letter 1]

22 Aug 2022

Challenges with pediatric surgical financing and universal health coverage in Guatemala: A qualitative analysis

PGPH-D-21-00647R1

Dear Dr Landrum,

We are pleased to inform you that your manuscript 'Challenges with pediatric surgical financing and universal health coverage in Guatemala: A qualitative analysis' has been provisionally accepted for publication in PLOS Global Public Health.

Best regards,

Augustine D. Asante

Academic Editor